# Glycan-Adhering Lectins and Experimental Evaluation of a Lectin FimH Inhibitor in Enterohemorrhagic *Escherichia coli* (EHEC) O157:H7 Strain EDL933

**DOI:** 10.3390/ijms23179931

**Published:** 2022-09-01

**Authors:** Jun-Young Park, Cheorl-Ho Kim, Seung-Hak Cho

**Affiliations:** 1Division of Zoonotic and Vector Borne Disease Research, Center for Infectious Disease Research, Korea National Institute of Health, Cheongju 28159, Korea; 2Glycobiology Unit, Department of Biological Science, Sung Kyunkwan University and Samsung Advanced Institute for Health Science and Technology (SAIHST), Suwon 16419, Korea

**Keywords:** enterohemorrhagic *Escherichia coli* (EHEC), lectin, glycan, FimH, Gb3

## Abstract

In this study, we tried to develop a FimH inhibitor that inhibits adhesion of enterohemorrhagic *Escherichia coli* (EHEC) on the epithelium of human intestine during the initial stage of infections. Using a T7 phage display method with a reference strain, EHEC EDL933, FimH was selected as an adherent lectin to GM1a and Gb3 glycans. In order to detect the ligand binding domain (LBD) of FimH, we used a docking simulation and found three binding site sequences of FimH, i.e., P1, P2, and P3. Among Gb3 mimic peptides, P2 was found to have the strongest binding strength. Moreover, in vitro treatment with peptide P2 inhibited binding activity in a concentration-dependent manner. Furthermore, we conducted confirmation experiments through several strains isolated from patients in Korea, EHEC NCCP15736, NCCP15737, and NCCP15739. In addition, we analyzed the evolutionary characteristics of the predicted FimH lectin-like adhesins to construct a lectin-glycan interaction (LGI). We selected 70 recently differentiated strains from the phylogenetic tree of 2240 strains with Shiga toxin in their genome. We can infer EHEC strains dynamically evolved but FimH was conserved during the evolution time according to the phylogenetic tree. Furthermore, FimH could be a reliable candidate of drug target in terms of evolution. We examined how pathogen lectins interact with host glycans early in infection in EDL933 as well as several field strains and confirmed that glycan-like peptides worked as an initial infection inhibitor.

## 1. Introduction

A class of virulence proteins, called lectin-like virulence proteins, is responsible for the pathogenesis of bacterial infections, and it may be exploited as a therapeutic target or vaccine components [1]. Bacterial adhesins are lectin proteins that have the ability to adhere to host cells and they have a variety of structural topologies [2]. Pili and fimbriae are some of the structures found in bacteria. These bind to host cell surface receptor proteins and participate in a variety of biological processes, including cross-membrane tracking, invasion, and migration [3]. They can be toxic to the host, causing inflammation [4]. Mannose supplementation or receptor inhibition may affect the adhesin–receptor interaction in certain adhesins, such as mannose, which is critical in immunological activation. The PilA glycoprotein from *Acinetobacter baumannii* binds to host cell selectins and carcinoembryonic antigen-related cell adhesion molecules (CEACAMs) [5]. Other lectin-like proteins include: *Escherichia coli’s* surface antigen 20 (CS20) and fimbriae protein SfaS; Hemophilus influenzae’s surface-adhesin protein E; Neisseria meningitidis’s autotransporter adhesin; and *Salmonella enterica* serovar Enteritidis’s ShdA, MisL, Sad, and BapA [6]. These proteins adhere to the host gut by the binding of lectin-like adhesins to host cell receptors that comprise glycans, resulting in lectin-glycan interactions (LGIs) [7].

*E. coli* is rod-shaped Gram-negative bacterium that is often found in the lower intestines of warm-blooded organisms, including humans [8]. Most *E. coli* serotypes are non-pathogenic, but some serotypes can cause food poisoning, such as enterohemorrhagic *Escherichia coli* (EHEC). One of the major serotypes of this group is O157:H7. Infection with this type of pathogenic *E. coli* can lead to hemorrhagic colitis or hemolytic uremic syndrome (HUS) [9,10]. Since the first confirmation of O157:H7 as a human pathogen in 1982, sporadic outbreaks of its infection have been reported in various regions of the world. There has been a long-term epidemic in Japan, and this strain has been recognized as an epidemiologically important infectious pathogen [11]. The Shiga-like toxins Stx1 and Stx2 are expressed in *E. coli* O157:H7. The genes encoding these toxins may be horizontally transmitted to *E. coli* or other *Enterobacteriaceae* species through prophage [12], enabling the transformation of non-Shiga-like toxin-generating strains into Shiga-like toxin producing strains [13]. Shiga toxins produced by Shiga toxin-producing *E. coli* (STEC) are mostly directed at capillary endothelial cells. Shiga toxins specifically target the globotriaosylceramide receptor on cells, and bacteria enter cells through receptor-mediated endocytosis [14]. Shiga toxins prevent protein synthesis by cleaving an adenine base from infected cells’ ribosomes [15]. As demonstrated in HUS [14], this obstruction may result in renal failure.

The majority of Gram-negative bacteria produce adhesins, and in many instances, the adhesin’s activity is mediated by a tiny component protein located at the tip of each *fimbriae*. For example, the bacterial adhesin FimH is responsible for the adhesion of *E. coli* to host intestinal cells via the activation of their glycan receptors, such as CD48 and TLR4, or via binding to mannose residues. Various attempts have been made to produce anti-adhesion vaccines, including those based on anti-FimH antibodies. Passive immunization with anti-FimH antibodies has been demonstrated to significantly reduce uropathogenic *E. coli* colonization in animal models [16]. On the contrary, the lectin-like adhesins of *E. coli* are not currently catalogued, making their subsequent use in vaccine development more difficult. As a result, further exploration of *E. coli* adhesins at the genome-wide level is desirable.

We have investigated how EHEC lectins interact with host glycans in early infection using a reference strain EHEC EDL933 as well as several field strains using the T7 phage display method. Furthermore, we attempted to develop FimH binding inhibitors in order to facilitate the discovery of therapeutics for the prevention of EHEC adherence during the initial stage of infections by using docking simulation, binding, and inhibition tests in EDL933 as well as several strains isolated from diarrheal patients in Korea (NCCP15736, NCCP15737, and NCCP15739). Additionally, we compared the genomes of EDL933 and field strains in order to examine their evolution and phylogenetic relationship [17]. Our results suggest that the peptide could play an important role in the development of therapeutic agents inhibiting the adherence of EHEC to epithelial cells of the human gastrointestinal tract.

## 2. Results

### 2.1. Finding Lectin Receptor of EDL933 Linked to Gb3 and GM1a Glycan Using T7 Phage Display

For the T7 phage display, mRNA was extracted from the strain EDL933. cDNA was synthesized with the extracted mRNA and a phage library was created. The detailed process is described in Materials and Methods. cDNA synthesis using oligo (dT), random, or the matching subunit-specific primers yielded comparable quantities of transcripts. The concentration and purity of the synthesized cDNA were confirmed by absorbance measurement and electrophoresis (Figure 1A).

The plaque lift assay revealed lectin factors linked to Gb3 and GM1a glycan. Thick spots were selected and sequenced. As a result, FimH lectin was mainly detected, and a subsequent experiment was conducted with FimH lectin (Figure 1B,C)**.**

### 2.2. Determination of Ligand-Binding Domain (LBD) of FimH

Docking simulations were used to determine the ligand-binding domain (LBD) of FimH that binds to Gb3 and GM1a glycans. We chose residues with a binding affinity of −3 kcal/mol or less for the FimH lectin residue interacting with the ligand, and −3 kcal/mol or less for other ligands, to locate the ligand-binding domain (LBD) of FimH. In addition, the binding strengths of GM1 and Gb3 to FimH lectins were higher for Gb3 than for GM1a. Three LBDs at the anticipated binding locations were discovered as a result (Figure 2A,B). Therefore, FimH was analyzed as a possible lectin receptor.

### 2.3. Comparison of the Adhesion Affinity and Inhibitory Effect of the Peptides of FimH LBDs Using EDL933

From our analysis, three LBDs were identified. The peptide candidate sequences for the binding site are mannose-6-phosphonate-conjugated CGTVLTRNETHATYS, CQCKQDFNITDISLL, and CYATPSSNATDPLKY, which were named P1, P2, and P3, respectively. An adhesion test with FimH lectin of EDL933 strain was conducted. The difference in adhesion affinity between the glycan mimic-peptides and the original glycan gb3 was compared. There was no significant difference in binding strength between the glycan gb3 and lectin FimH of EDL933. Among the glycan mimic peptides, P2 was found to have the strongest binding strength. Compared with the control, P1, P2, and P3 showed significant variations in their capacity to bind to FimH (Figure 3A). We speculated that the Gb3 glycan-like peptide may bind to FimH and affect its binding to host glycans. Moreover, as shown in Figure 3B, in vitro treatment with peptide P2 inhibited binding activity in a concentration-dependent manner. EDL933 were treated with peptide P2 at 0, 10, 100, and 1000 µM. The higher the concentration in the EDL933 strain, the more inhibited it was, according to the result data.

### 2.4. Confirmation of the Inhibition Affinity of the Peptides of FimH LBDs Using the Stains Isolated from Patients

We conducted confirmation experiments through several field strains of EHEC NCCP15736, NCCP15737, and NCCP15739 as well as EDL933. An adhesion test with FimH lectin of the separating strain was conducted. P2 was also found to have the strongest binding strength. Compared with the control, P1, P2, and P3 showed significant variations in their capacity to bind to FimH (Figure 4A). Moreover, as shown in Figure 4B, in vitro treatment with peptide P2 inhibited binding activity in a concentration-dependent manner. NCCP15736, NCCP15737, and NCCP15739 were treated with peptide P2 at 0, 10, 100, and 1000 µM. As shown in the results from EDL933 strain, the higher the concentration, the more inhibiting results were confirmed.

### 2.5. Evolutionary Analysis of EHEC Strains and Gene-Based Subtyping of EHEC Strains in Relation to LGI

To determine the complete evolutionary history of EHEC strains, we conducted a multiple sequence alignment analysis of 2204 STEC strains (Appendix A) and *S. flexneri* 2a str. 2457T. From the NJ tree of 2204 STEC strains, we selected 70 strains (Appendix A) that represent strains recently evolved to assess how many evolutionary events are affected in their genomic evolution (Figure 5A, Appendix A). Multi-locus sequence typing (MLST)-based phylogenetic analysis revealed that a NCCP strain, NCCP15739, previously reported by us was near EDL933, but NCCP15736 and NCCP15737 were distant from EDL933 in the phylogenetic tree.

Owing to the vast diversity of *E. coli*, reliance on gene-based subtyping may result in the identification of new groups of closely related isolates as a result of gene variation or links among unrelated isolates due to recombination. Moreover, we used a gene-based technique to classify the genomes into lineages of closely related isolates. For the gene-based subtyping, we selected a gene, FimH, as the gene encodes a major adhesin in EHEC strains. Based on the gene-based subtyping described above, the grouping produced one major lineage (Figure 5B). It was confirmed that not only EDL933 but also other EHEC strains, which are reference strains, are associated with FimH. We can infer EHEC strains were dynamically differentiated but FimH was conserved during the evolution time according to the phylogenetic tree. The experimental results also show that isolated strains have the same pattern as standard strains. Because of the evolutionary conservation, FimH could be a reliable candidate of drug target in terms of evolution.

## 3. Discussion

Antibiotic treatment of patients infected with EHEC is not recommended because EHEC could increase Shiga toxin production and release in response to antibiotics, which eventually increases the frequency and severity of clinical symptoms [18,19,20]. There is currently no treatment for EHEC infection. Therefore, we designed and conducted this study for the development of moderate therapeutics against EHEC. Both non-experimental and experimental methodologies are required to predict lectin candidate proteins in pathogenic bacteria. FimH, which is a mannose-specific adhesin that mediates shear-enhanced bacterial adherence and is found at the tip of type 1 fimbriae of *E. coli*, plays an important role in uropathogenic *E. coli* (UPEC) for the binding and colonization of urothelial cells (uroplakins), as well as UPEC invasion of bladder epithelial cells [21]. FimH facilitates bacterial attachment to urothelial cells, which is the first stage in the infection process [22]. It is made up of two domains, a pilin domain associated with the fimbria and a mannose-binding lectin domain, with the binding pocket on the other side of the interdomain interface [23]. The anti-FimH activity of several mannose derivatives has been shown and is, consequently, regarded to be potential therapeutic agents for the treatment of urinary tract infections [24,25]. However, in the case of EHEC and ETEC, the binding to FimH is not well understood. Thus, it is important to investigate the role of FimH in EHECs.

Several studies have elucidated the interaction between pathogen lectins and host glycans during the early stages of infection [6,7,26]. We established an experimental strategy and method to characterize the systemic LGI of pathogenic intestinal bacteria in the gastrointestinal tract. We used the T7 phage display method to identify binding peptides or proteins to glycans [27,28,29]. Plaques with a high concentration of GM1a and Gb3 were found to be associated with intestinal epithelial cells; thus, novel FimH-related lectins were discovered in EDL933.

Furthermore, we conducted various experiments to analyze FimH inhibitors using NCCP15736, NCCP15737, and NCCP15739 strains as field isolates can be more useful for the discovery of new medical agents than reference strains such as EDL933. Moreover, genome-based screening is a non-experimental technique to identify lectins from pathogenic bacteria [6]. EHEC strains are recognized as epidemiologically important infectious pathogens as sporadic outbreaks of their infections can result in an epidemic [30,31]. Therefore, it is important to perform genomic sequencing and investigate their evolution [32,33]. Moreover, the analysis of gene-based subtyping using the FimH gene suggested that field strains belong to the large group of EDL933. From these results, it can be assumed that not only the EDL933 reference strain, but also other field strains, have a general mechanism of FimH lectin-glycan interaction.

We then investigated the ability of the glycan-like peptides to examine whether they reduce or prevent early pathogenic infection via binding to FimH lectin using a reference strain and patient isolates. The results help to determine their clinical applicability in the prevention of EHEC infection. Therefore, the adsorption/removal ability, functional inhibitory activity, and in vitro effect of the obtained glycan-like peptides were investigated. In the case of the P2 mimetic peptide, the infection inhibitory effect was confirmed. The current investigation has demonstrated the efficacy of glycan-mimic peptides to disrupt binding to lectin. This is considered as a novel therapeutic strategy for the prevention and treatment of intestinal infections for which there is yet no effective treatment because antibiotics rather induce production of pathogenic toxins [18,19,20]. However, there are still several barriers to the successful prevention and treatment of EHEC infection due to the pathogenic complexity of EHEC infection, which includes disruption of key homeostatic pathways involving in complex biochemical and physiological systems [34]. We are, therefore, exploring the possibility of testing bioactive peptides for the prevention of early EHEC infection in both intestinal epithelial cell models and animal models to further enhance clinical applicability [35].

Additionally, we demonstrated that Gb3 had a better binding capability to FimH than GM1a and used simulated docking to determine the binding site sequence. Docking modeling was used to predict the structures of sugar chains that may bind to lectin ligands. These protein-ligand and sugar chain-lectin ligand docking approaches were used to estimate the binding structure of ligands. Additionally, these strategies were used to identify and optimize binding sites and candidate materials [6,7,26].

The array technology, which is used to assess the specificity of the pathogen lectin for the host glycan, can be used to evaluate hundreds of samples concurrently; moreover, it requires only a small quantity of each sample [36]. By using qualitative or quantitative assessments of the specificity of the pathogen lectin for the host glycan, this approach may be utilized to identify a likely host glycoprotein that binds to an adhesin of a pathogen [37]. Here, we sought to determine whether the adhesin was bound using the three Gb3 attachment peptide candidate groups P1, P2, and P3 and found that P2 has the highest binding affinity. These results suggest that the peptide may act as an important inhibitor restraining the adherence of EHEC to epithelial cells of the human gastrointestinal tract during the initial stage of EHEC infection (Figure 6). However, more experiments for the confirmation of binding capacity using cell lines or animal models are needed. The protein sequences and structures of pathogen lectins may aid the development of a vaccine or antibiotic substitute for therapeutic usage.

## 4. Materials and Methods

### 4.1. Strain, Isolation, and Serotyping

A reference strain and three field strains were used. The strains were grown in LB broth overnight at 37 °C. EDL933 was used as the reference strain. In addition, we included previously published EHEC strains that were isolated from diarrheal patients in Korea: NCCP15736 [38], NCCP15737 [38], and NCCP15739 [39] for the experiments and comparative genomic analysis (Table 1).

### 4.2. Bacterial Growth Conditions

The strain EHEC EDL933 was cultured on a solid LB plate for harvesting an appropriate amount of strain, and the amount was compared by measuring the titer and absorbance of the liquid LB medium. It was confirmed that the absorbance 0.1 at 600 nm of the liquid strain culture was equivalent to 1.6 × 10^8^ cells/mL.

### 4.3. Total RNA Isolation

It was decided to extract RNA from a strain of mid-log phase of EDL933, known to have the highest RNA expression level, harvested and cultured with shaking at 37 °C and 220–250 rpm for 2–3 h. Then, 5 × 10^8^ to 1 × 10^10^ strains of mid-log phage were harvested and total RNA was extracted according to the enzymatic lysis protocol of the RNeasy Midi Kit (Qiagen, Hilden, Germany) [40]. cDNA synthesis was performed with RNA having a concentration of 1.0 μg/μL or more and an A260/A280 absorbance ratio of 1.9 to 2.1.

### 4.4. Synthesis of Double-Stranded cDNA through Reverse Transcription PCR

Using the total RNA of EDL933 as a template, according to the protocol of the SuperScript Double-Stranded cDNA Synthesis Kit (invitrogen, Waltham, MA, USA), double-stranded cDNA was synthesized through reverse transcription PCR and purified by phenol extraction [41]. Random hexamer: d(NNNNNN) primer was used to synthesize the T7 phage and cDNA to be inserted.

### 4.5. T7 Phage Display

A phage library was prepared according to the (Novagen, Madison, WI, USA) protocol of the T7Select phage system. A custom-made Directional EcoR I/Hind III linker: d (GCTTGAATTCAAGC) was ligated with T4 DNA ligase (invitrogen) at both ends of the smoothed cDNA with 16 h at 16 °C overnight ligation or according to the manufacturer’s protocol and purified using the kit or phenol extraction. The cDNA-linker ligated product was reacted with FastDigest EcoR I and Hind III (invitrogen) restriction enzymes at 37 °C for 30 min to make aromatic sticky-ends and were purified using the kit or phenol extraction. Restriction enzyme-treated cDNA fragment was inserted into T7 vector, and restriction enzyme-treated cDNA fragment was subjected to sticky-end ligation to T7Select 10-3b vector (Novagen). The manufacturer’s protocol for T4 DNA ligase (invitrogen) was followed. The prepared phage vector library was reacted with T7Select packaging extract for 2 h at 22 °C to prepare two types of T7 phage library and was infected with the host *E. coli* strain BLT5403 and amplified [42]. As an antibiotic, 50 ug/mL of Carbenicillin was used. After confirming the titer of the primary amplified phages through plaque assay, they were refrigerated or cryopreserved until biopanning.

### 4.6. Plaque Lift Assay

EHEC bacteria in log phase growth were infected with various dilutions of phage and grown on agar plates. About 1000 well-dispersed individual phage clones from agar plates were transferred to nitrocellulose membranes using the plaque-lift technique. A nitrocellulose membrane (82-mm diameter HATF filter, Millipore Corp., Burlington, MA, USA) was overlaid onto the agar and incubated at room temperature for 10–15 min. Without disrupting the agar and plaques, the nitrocellulose membrane was carefully lifted. The immunostaining procedure included an initial incubation with the binding glycan (GM1 and Gb3, respectively)-EHEC for 1–2 h. The membranes were then rinsed, followed by the application of detecting reagents. The detecting reagents include horseradish peroxidase-conjugated goat anti-mouse IgG, followed by color development using 3,3-diaminobenzidine (DAB) and 0.01% H_2_O_2_.

### 4.7. Docking Simulation

Furthermore, this method enabled us to decipher the structure of the pathogen’s lectin and to identify the site of attachment to the host’s receptor. Using SWISS-MODEL [43], a homology modeling approach, the three-dimensional (3D) structure of FimH (PDB ID: 6GTY and Sequence:MKRVITLFAVLLMGWSVNAWSFACKTANGTAIPIGGGSANVYVNLAPAVNVGQNLVVDLSTQIFCHNDYPETITDYVTLQRGAAYGGVLSSFSGTVKYNGSSYPFPTTSETPRVVYNSRTDKPWPVALYLTPVSSAGGVAIKAGSLIAVLILRQTNNYNSDDFQFVWNIYANNDVVVPTGGCDVSARDVTVTLPDYPGSVPIPLTVYCAKSQNLGYYLSGTTADAGNSIFTNTASFSPAQGVGVQLTRNGTIIPANNTVSLGAVGTSAVSLGLTANYARTGGQVTAGNVQSIIGVTFVYQ) was produced using GM1(PDB ID: 4ZH1)- and Gb3(PDB ID: 6F4C) 3D structure. The docking simulation was conducted using the 3D structure of GM1 and Gb3. The Chem-office application (http://www.cambridgesoft.com, version: 7.0, accessed on 8 July 2022) was also used to reduce energy use. Autodock Vina 1.1.2 was used for the docking simulations [44]. Possible hydrogen bonds and hydrophobic interactions were identified utilizing HBPLUS and non-bonded contact parameters as default settings in a LigPlot based on the findings of docking simulation [45].

### 4.8. Solid Phase Peptide Synthesis

The initial step in synthesizing peptides on a resin was to connect the C-terminal amino acid to the resin. A transient protective group protects the alpha amino group and the reactive side chains from polymerization. The resin was next filtered and washed to eliminate byproducts and excess reagents. The N-alpha protective group was then removed, and the resin was washed again to eliminate byproducts and excess reagent. Then, the next amino acid was added until the peptide sequence was complete. It was then rinsed to remove the protective groups and the peptide was released from the resin. As in the above procedure, the peptide manufacturing synthesis proceeded with the same protocol for the mannose-6-phosphonate conjugated P1, P2, and P3. The Gb3-like peptides, P1 (CGTVLTRNETHATYS), P2 (CQCKQDFNITDISLL), and P3 (CYATPSSNATDPLKY) were similarly synthesized.

### 4.9. Peptide Binding and Inhibition Assays

The mannose-6-phosphonate-conjugated P1, P2, and P3 peptides were coated to each well of the plates at a concentration of 25 ng per well. After blocking, each well was filled with EHEC for 2 h at room temperature and washed five times with 1% BSA. Next, the plate was incubated at 4 °C for 15–16 h. After another wash, the plate was stained with fluorescein isothiocyanate-conjugated staining reagent at a 1:10,000 dilution. After 2 h at room temperature, the plates were washed, followed by fluorescence detection using a micro reader.

### 4.10. Phylogenetic Analysis and Comparative Genomic Analysis

A total of 20,603 *E. coli* genomes were downloaded from the National Center for Biotechnology Information (NCBI) and curated to create a final collection of 2204 EHEC genomes with Shiga toxin genes. To infer the evolutionary history of field strains, we performed a multiple sequence alignment of the MLST genes using MEGA X (version 10.0.5) [46]. A MUSCLE algorithm was used for multiple sequence sorting, and the neighbor joining (NJ) method [47] was used as the clustering algorithm. From the NJ tree of 2204 STEC strains, we selected 70 strains that represent recently evolved strains. Multiple sequence alignment results were saved in Mega format. The unweighted pair group method with arithmetic mean (UPGMA) was used as a statistical test method for estimating phylogenetic trees. The phylogenetic tree test was performed using the bootstrap method, with 1000 iterations. The substitution model used maximum composite likelihood and included both transitions and transversions. Calculation results were saved in nexus format. The tree was visualized with FigTree (version 1.3.1) (http://tree.bio.ed.ac.uk/software/figtree/, accessed on 20 July 2022). In order to exclude the effect of HGT in our phylogenetic analysis, we used the multi-locus sequence analysis method [48,49]. Seven housekeeping genes (*adk*, *fumC*, *gyrB*, *icd*, *mdh*, *purA*, and *recA)* from 70 *E. coli* strains were retrieved and concatenated. A phylogenetic tree of MLST genes was created using the method employed for MLST-based phylogenetic analysis [50].

### 4.11. Gene-Based Subtyping of EHEC Strains in Relation to LGI

During infection, EHEC strains attach to the host intestinal cell by the binding of adhesins to receptors of the host, such as glycans. Bacterial adhesins, such as lectin, are potential therapeutic targets and can contribute to an understanding of bacterial evolution. In our previous study, we constructed the LGI network of EHEC [6] and developed an e-Membranome database [26]. Using an e-Membranome pipeline, we predicted putative adhesins from EHEC genomes and deposited them onto a public database. Among the putative adhesins, we selected the FimH gene and performed gene-based subtyping of EHEC strains. The nucleotide sequences of FimH gene were retrieved from 70 EHEC strains. We performed a phylogenetic analysis of the FimH gene using MEGA X (version 10.0.5), as described above.

### 4.12. Data Accuracy Treatment

For accuracy of the obtained results shown in Figure 1B,C, the plaque lift assays were repeated at least three times to visualize the bound dots using the EDL933 Gb3 and GM1a glycans. In addition, the binding affinities and inhibitory effects shown in Figure 3A,B and Figure 4A,B have been obtained from the repeated measurements at least three times. The results are subjected to statistical analysis.

### 4.13. Statistical Analysis

All experiments were carried out at least three times, and representative results are shown. The outcomes of the data analysis were statistically analyzed using the comparison-based one-way analysis of variance (ANOVA), which was then followed by a post hoc Bonferoni test to determine significance. Differences were considered statistically significant when their *p*-values were less than 0.05. * *p* indicates < 0.05 and ** *p* < 0.01. The differences between the two figures are indicated in the figure legends.

## 5. Conclusions

We examined how pathogen lectins interact with host glycans early in infection in EDL933 as well as several separate strains (NCCP15736, NCCP15737, and NCCP15739). Binding of the lectin FimH to the intestinal host glycan was confirmed not only in the EDL933 strain but also in other EHEC strains. In addition, the effect of synthesizing glycan-like peptides and inhibiting them was tested. Through LGI studies, we recommend further studies on lectins and glycans to lay the groundwork for developing therapeutic methods for gastrointestinal infections.

## Figures and Tables

**Figure 1 ijms-23-09931-f001:**
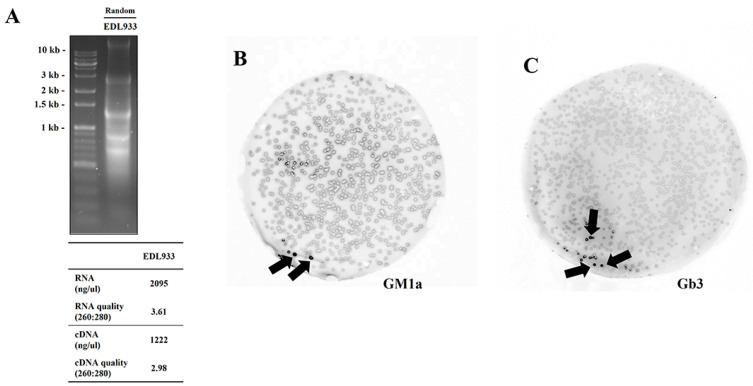
cDNA synthesis and the plaque lift assay revealed lectin factors linked to Gb3 and GM1a glycan in EDL933. (**A**) cDNA synthesis. (**B**,**C**) The lectin factors associated with Gb3 and GM1a glycans were identified using a plaque lift assay.

**Figure 2 ijms-23-09931-f002:**
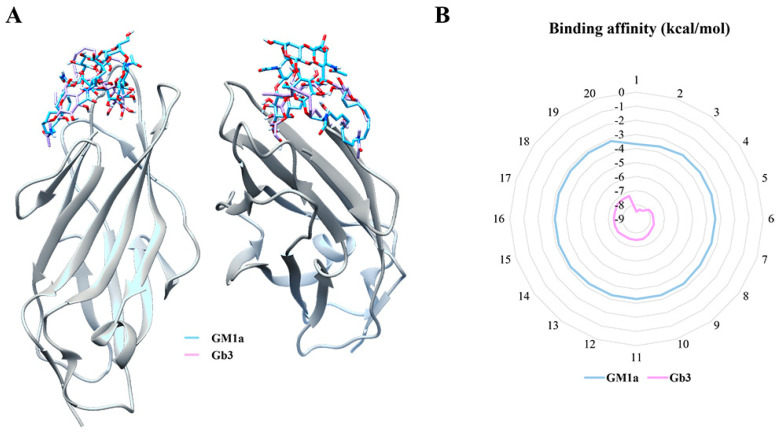
Docking simulations with FimH, a lectin candidate that binds to Gb3 and GM1a glycans, were used to determine binding affinity. (**A**) Binding affinity was determined using docking simulations with FimH. (**B**) The binding strength (kcal/mol) to FimH lectins was stronger for Gb3 than for GM1a, and three LBDs at the expected binding sites were discovered.

**Figure 3 ijms-23-09931-f003:**
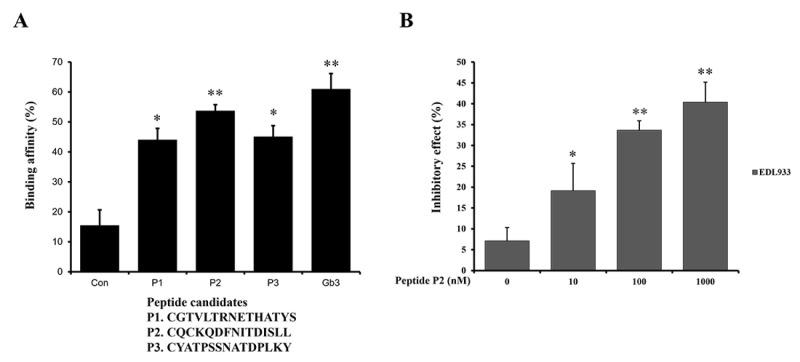
Elucidation of inhibitor restraining linkage of EDL933 strain. (**A**) A comparison of the binding affinities of Gb3-replica peptides to FimH in EDL933 was made. The term “control” refers to the binding affinity of a matrix without peptides. When compared to the control, the P1–P3 peptides revealed statistically significant changes. (**B**) In vitro treatment with peptide P2 inhibited binding activity in a concentration-dependent manner. EDL933 were treated with compounds at 0, 10, 100, and 1000 nM. When compared to the control, the P1–P3 peptides revealed statistically significant changes. * Means were substantially different (* *p* < 0.05 and ** *p* < 0.01).

**Figure 4 ijms-23-09931-f004:**
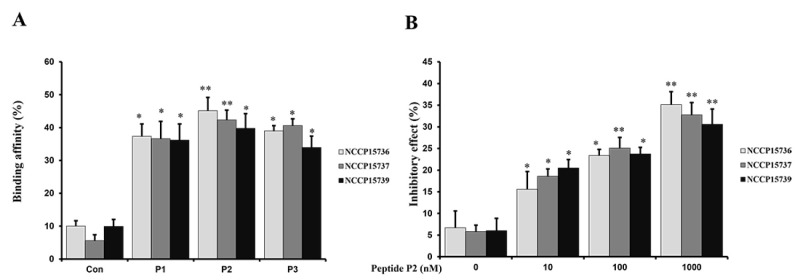
Elucidation of inhibitor restraining linkage of pathogenic enteric bacteria EHEC stains (NCCP15736, NCCP15737, and NCCP15739). (**A**) A comparison of the binding affinities of Gb3-replica peptides to FimH in EHEC stains (NCCP15736, NCCP15737, and NCCP15739) was made. The term “control” refers to the binding affinity of a matrix without peptides. When compared to the control, the P1–P3 peptides revealed statistically significant changes. (**B**) In vitro treatment with peptide P2 inhibited binding activity in a concentration-dependent manner. NCCP15736, NCCP15737, and NCCP15739 were treated with compounds at 0, 10, 100, and 1000 nM. When compared to the control, the P2 peptides revealed statistically significant changes. * Means were substantially different (* *p* < 0.05 and ** *p* < 0.01).

**Figure 5 ijms-23-09931-f005:**
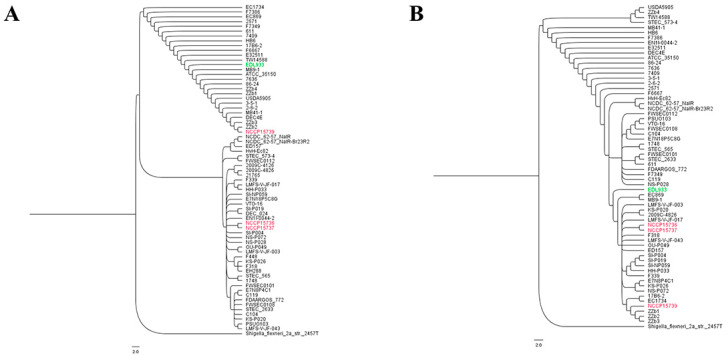
Multi-locus sequence typing (MLST)-based phylogenetic tree of EHEC strains. MLST-based phylogeny and phylogenetic tree of EHEC strains with respect to the FimH gene. (**A**) Evolutionary time scaled by 100; lower values imply relatively recent branching. The scale indicates the number of substitutions per site. (**B**) Evolutionary time scaled by 100; lower values imply relatively recent branching. The scale indicates the number of substitutions per site. Green: reference stain, Red: the stains isolated from patients.

**Figure 6 ijms-23-09931-f006:**
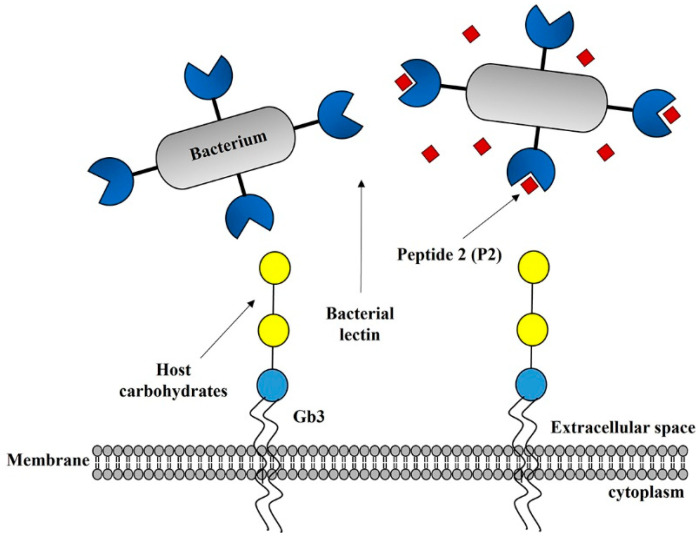
Diagrammatic illustration of elucidating an inhibitor-restraining connection in the EDL933 of pathogenic enteric bacteria. Immunotherapy for EHEC EDL933 might be used to prevent and cure infectious disorders by determining the involvement of the host Gb3 glycan peptide in the immune response against the pathogen.

**Table 1 ijms-23-09931-t001:** Strains used in this study.

Name	NCBI Nucleotide Accession	Assembly	References
NCCP15736	AOUQ00000000.1	GCA_000403965.1	Ref. [38]
NCCP15737	AOUP00000000.1	GCA_000403985.1	Ref. [38]
NCCP15739	ASHA00000000.1	GCA_000392555.1	Ref. [39]

## Data Availability

Not applicable.

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
