# Peer review of "Glycan-Adhering Lectins and Experimental Evaluation of a Lectin FimH Inhibitor in Enterohemorrhagic Escherichia coli (EHEC) O157:H7 Strain EDL933"

_ijms, 2022, doi:10.3390/ijms23179931_

Round 1
Reviewer 1 Report
Dear Authors,
The manuscript ID: ijms-1884210_v1 entitled „Glycan-Adhering Lectins and Experimental Evaluation of a Lectin FimH Inhibitor in Enterohemorrhagic Escherichia coli (EHEC) O157:H7 strain EDL933” written by Jun-Young Park, Cheorl-Ho Kim and Seung-Hak Cho is very interesting and topical.
The purpose of this study – an attempt to develop inhibitors of FimH binding to facilitate the discovery of therapeutic agents to prevent enterohemorrhagic Escherichia coli (EHEC) infections in the initial phase of infection is very original. In recent years, bacterial resistance to antimicrobials has become very widespread and is one of the most serious global threats to public health. Therefore, it is necessary to search for new antibacterial therapies, including the treatment of E. coli infections.
The whole manuscript is properly organized. Introduction contains general data on EHEC, which are responsible for bloody dysentery and an increased risk of hemolytic uremic syndrome. Lectin-like virulence proteins that can be used as a therapeutic target or vaccine components have also been described. Appropriate materials and methods were used to perform these studies with both reference and clinical strains. Very detailed research has been carried out. Statistical analyzes were also performed. The obtained results are documented, summarized in the form of figures or schemes and properly interpreted. Based on the results, discussion and conclusions were drawn. I agree with the Authors that the study of the interaction of EHEC lectins with host glycans at an early stage of infection and the evaluation of the effect of the synthesis of glycan-like peptides contributes a lot to the development of new therapeutic methods for gastrointestinal infections. These research is needed to prevent intestinal infection or treat it in its early stages.
It is a well written article. I have only small suggestions in order to improve paper, which are the following:
Line 39: Acinetobacter baumannii – italics;
Lines 127, 146: in vitro – italics;
Line 140: idolated – isolated
Line 283: (invitorgen) – (invitrogen)
Line 293: E. coli – italics;
Lines 392-393: in infection in infection – please correct (it is written twice)
In my opinion, these results are original, valuable and manuscript may be accepted and published in such a prestigious journal as “International Journal of Molecular Sciences”.
With highest regards,

Author Response
I appreciate the reviewer for careful reading.
Please see the attachment.

Reviewer 2 Report
Introduction
This is OK in general, although some of the points presented can also be used in the discussion to explain some of the findings.
At the end of the session, please summarise the hypothesis of the research and also describe clearly the objectives of the study.
Methodology
This is ok, but the bioinformatics part lacks the references for all the work performed. These must be added.
Analysis. 1) how did you treat results of repeated measures? 2) can you confirm that results had a normal distribution?
Results
I want to see all the results of the bioinformatics analysis in supplementary material. Please start with presentation of the 70 strains and proceed along. This is serious omission that will lead directly to rejection if not corrected, as it raised significant questions.
If the authors have the results, they must provide all the details.
Discussion
Please add a new paragraph regarding the clinical implications of the findings. Also, what do the findings tell us about the immune response of patients?
The work needs to be corrected as detailed above. As it is now, it cannot be published.
Author Response

(The authors gave the same response as above.)

Round 2
Reviewer 2 Report
The authors did not address correctly how they dealt with the repeated measurements of their tests.
This must be corrected before final acceptance.
Author Response
Minor Revision
Comments and Suggestions for Authors
The authors did not address correctly how they dealt with the repeated measurements of their tests.
This must be corrected before final acceptance.
The answer: I appreciate the reviewer for careful reading. I am sorry for our previous mistakes. As suggested, the Data accuracy treatment of Methodology section have been added.
“Data accuracy treatment :
For accuracy of the obtained results shown in Fig.1B and Fig.1C, the plaque lift assays were repeated at least three times to visualize the bound dots using the EDL933 Gb3 and GM1a glycans. In addition, the binding affinities and inhibitory effects shown in Fig. 3A, Fig. 3B, Fig.4A and Fig 4B have been obtained from the repeated measurements at least 3 times. The results are subjected to statistical analysis.”
Also, the revised manuscript is attached.
